# Immobilization of Alcalase on Silica Supports Modified with Carbosilane and PAMAM Dendrimers

**DOI:** 10.3390/ijms232416102

**Published:** 2022-12-17

**Authors:** María Sánchez-Milla, Ester Hernández-Corroto, Javier Sánchez-Nieves, Rafael Gómez, María Luisa Marina, María Concepción García, F. Javier de la Mata

**Affiliations:** 1Departamento de Química Orgánica y Química Inorgánica, Universidad de Alcalá, Ctra. Madrid-Barcelona Km. 33.600, Alcalá de Henares, 28871 Madrid, Spain; 2Networking Research Center on Bioengineering, Biomaterials and Nanomedicine (CIBER-BBN), 28029 Madrid, Spain; 3Instituto de Investigación Química “Andrés M. del Río”, Ctra. Madrid-Barcelona Km. 33.600, Universidad de Alcalá, Alcalá de Henares, 28871 Madrid, Spain; 4Departamento de Química Analítica, Química Física e Ingeniería Química, Ctra. Madrid-Barcelona Km. 33.600, Universidad de Alcalá, Alcalá de Henares, 28871 Madrid, Spain; 5Instituto Ramón y Cajal de Investigación Sanitaria (IRYCIS), Ctra. de Colmenar Viejo, Km. 9,100 Planta – 2 derecha, 28034 Madrid, Spain

**Keywords:** enzyme immobilization, alcalase, silica, carbosilane dendrimer, PAMAM dendrimer

## Abstract

Enzyme immobilization is a powerful strategy for enzyme stabilization and recyclability. Materials covered with multipoint molecules are very attractive for this goal, since the number of active moieties to attach the enzyme increases with respect to monofunctional linkers. This work evaluates different dendrimers supported on silica to immobilize a protease enzyme, Alcalase. Five different dendrimers were employed: two carbosilane (CBS) dendrimers of different generations (SiO_2_-G_0_Si-NH_2_ and SiO_2_-G_1_Si-NH_2_), a CBS dendrimer with a polyphenoxo core (SiO_2_-G_1_O_3_-NH_2_), and two commercial polyamidoamine (PAMAM) dendrimers of different generations (SiO_2_-G_0_PAMAM-NH_2_ and SiO_2_-G_1_PAMAM-NH_2_). The results were compared with a silica support modified with a monofunctional molecule (2-aminoethanethiol). The effect of the dendrimer generation, the immobilization conditions (immobilization time, Alcalase/SiO_2_ ratio, and presence of Ca^2+^ ions), and the digestion conditions (temperature, time, amount of support, and stirring speed) on Alcalase activity has been evaluated. Enzyme immobilization and its activity were highly affected by the kind of dendrimer and its generation, observing the most favorable behavior with SiO_2_-G_0_PAMAM-NH_2_. The enzyme immobilized on this support was used in two consecutive digestions and, unlike CBS supports, it did not retain peptides released in the digestion.

## 1. Introduction

The unique catalytic properties of enzymes have made them widely used in industrial and laboratory processes. Among them, proteases capture 60% of the enzyme market of the food industry, in addition to other uses in fine chemistry and within the detergent industry [1,2]. The huge application of proteases, its high cost, and the search for a more sustainable use of these molecules have boosted much research targeting its reusability. However, their protein nature, responsible for these attractive properties, is also responsible of their instability.

To counteract these problems, enzyme immobilization has been proposed as an alternative [2,3]. This strategy has been effective on increasing stability and favoring recyclability, which opens the possibility to employ them in continuous processes, although it usually reduces activity. Polymers, resins, membranes, silica, and nanoparticles have been employed as supporting materials for this immobilization [4,5]. Immobilization can be performed through different procedures [6,7], such as, for example, chemical bonding using reactive moieties of the enzyme (-NH_2_, -CO_2_H, -SH, -OH). When choosing the reactive group, it is important that the active region is not affected. Additionally, to favor enzyme anchoring to a surface, this surface has to be modified with high reactive groups such as aldehydes, epoxides, vinyl sulfone, etc., since these groups lead to softer immobilization conditions [6,8,9,10].

Another relevant factor to take into account in enzyme immobilization is the number of functions used for this process. Multipoint attachment increases enzyme rigidity and also stability. However, an adequate balance is necessary to keep activity as high as possible, since multipoint attachment does not always favor activity [11,12,13]. A way to obtain this multipoint attachment is by introducing multifunction molecules on the surface where the enzyme will be immobilized: the higher the number of reactive points on the surface, the increased possibility to bind enzymes. Hence, the use of multivalent molecules as dendrimers is very attractive [14,15,16,17,18].

Dendrimers are hyperbranched molecules with a well-defined structure that are synthesized step-by-step from a core, leading to a multifunctional surface [19,20,21,22]. Due to this surface, dendrimers can be used as linkers between a solid surface and the active molecules for their immobilization, such as, for example, enzymes. Previous results, reported in the bibliography, have shown the usefulness of enzyme-anchoring through dendrimers to prepare sensors, improving sensibility and selectivity [23].

Our group has developed carbosilane (CBS) dendritic systems [24,25,26,27] that have been successfully bound to different material surfaces [28,29,30,31], including a silica surface [32,33], exposing active functions for further applications. Recently, our group has employed these supports for the immobilization of thermolysin, an enzyme with specificity towards peptide bonds within hydrophobic residues (leucine, phenylalanine, valine, isoleucine, alanine, and methionine). The results demonstrated that G_0_ CBS dendrimers were more suitable for the immobilization of thermolysin than G_1_ ones, and that the immobilized thermolysin could be employed in three consecutive hydrolysis cycles [34].

Alcalase is an unspecific protease from *Bacillus licheniformis*, subtilisin Carlsberg, which is widely used by the food industry for the preparation of protein hydrolysates with low allergenicity and increased bioactivity. Alcalase has been immobilized on different supports: silica [35], chitosan [36,37], alginate beads [38], agarose beads [39], magnetic particles [40,41], glass sol-gel matrices [42], etc. Nevertheless, no work has evaluated the potential of dendrimer supports for the immobilization of this enzyme.

The aim of this work has been to immobilize Alcalase on silica using different dendrimer-based supports: CBS with a hydrophobic framework; and polyamidoamine (PAMAM) with a hydrophilic framework, with different generations (G_0_ and G_1_). Moreover, the influence of the immobilization and digestion conditions on the catalytic activity of the enzyme and the reusability of the immobilized enzyme were also evaluated. The results were of high interest, since there is no information about the influence of different multipoint linker structures on Alcalase activity.

## 2. Results and Discussion

### 2.1. Alcalase Immobilization

Taking into account previous results obtained with the thermolysin enzyme [34], Alcalase was immobilized on silica supports modified with a monofunctional ligand (SiO_2_-L-NH_2_; **1-SiO_2_**), CBS dendrimers of different generations (SiO_2_-G0Si-NH_2_, **2-SiO_2_**; SiO_2_-G_1_Si-NH2; **3-SiO_2_**) and core (SiO_2_-G_1_O_3_-NH_2_; **4-SiO_2_**), and PAMAM dendrimers of different generations (SiO_2_-G_0_PAMAM-NH_2_, **5-SiO_2_**; SiO_2_-G_1_PAMAM-NH_2_; **6-SiO_2_**) (Figure 1 and Appendix A). The silica support with the monofunctional molecule (**1-SiO_2_**) was used as control to evaluate the dendritic effect. Whereas **1-SiO_2_** only shows one amino group available for immobilizing the enzyme, **2-SiO_2_** and **5-SiO_2_** present three amino groups, **4-SiO_2_** shows four amino groups, and **3-SiO_2_** and **6-SiO_2_** show seven amino groups. The main difference between both types of dendritic frameworks is the hydrophobicity of CBS dendrimers and the hydrophilicity of PAMAM dendrimers. The modification of silica with these ligands was confirmed by comparison of the FTIR spectra of bared silica (SiO_2_) and modified supports (**X-SiO_2_**, X = 1–6) [34]. IR spectra of the bared silica support showed signals at 1036–1060 cm^−1^ and 784–796 cm^−1^, whereas modified supports showed additional broad bands at 1540–1660 cm^−1^, corresponding with C-H and N-H bending vibration.

Modified supports **X-SiO_2_** with amine functions were next employed in the immobilization of the Alcalase enzyme following the procedure described in Figure 2. These materials reacted with excess glutaraldehyde to generate aldehyde moieties on the silica surface. The amount of required glutaraldehyde, in every case, was estimated according to the number of amino groups of the supports. Thus, supports were activated with five moles of the glutaraldehyde/amino group to guarantee its efficient attachment. Previous works reported that a high concentration of glutaraldehyde allows two molecules of glutaraldehyde per amino group, which guarantees a high reactivity towards amino group in the enzyme [37]. Next, these aldehydes groups were used to anchor Alcalase by reaction with enzyme-NH_2_ groups (present in amino acids, lysine, arginine, asparagine, and glutamine and the terminal amino group).

In both previous steps, imine bonds were formed by the reaction between the primary amines and the aldehyde functions. These bonds can be hydrolyzed, leading to the release of the enzyme. To avoid this problem, excess NaHB_4_ was added to reduce reactive double bonds. The amount of NaHB_4_ was also estimated, taking into account the number of amino groups on the modified silica. At least three moles of NaHB_4_ were employed per amino group to guarantee the suitable reduction of bonds. 

After purification, the best procedure to determine the immobilization of enzyme anchored to the silicas was by hydrolyzing these new materials (**X-SiO_2_-Alc**) and analyzing the release of amino acids in this process. Figure 3 shows the amount of amino acids released in the hydrolysis of **X-SiO_2_-Alc** with hydrochloric acid. All the X-SiO_2_-Alc supports retained the enzyme in some extension. A higher amount of amino acids was observed for **5-SiO_2_-Alc**, which was modified with a PAMAM dendrimer. On the other hand, silicas modified with CBS dendrimers with a Si atom core, **2-SiO_2_-Alc** and **3-SiO_2_-Alc**, were the silicas with a smaller amount of Alcalase incorporated, even less than the monoligand silica, **1-SiO_2_-Alc**. Comparing with the results obtained in the immobilization of thermolysin using the same supports [34], it was observed that Alcalase was more retained in PAMAM supports than thermolysin.

Finally, the morphology of these new silicas was studied by SEM (Figure 4). All supports showed amorphous morphology, and no significant difference was observed between those attaching Alcalase and those without the enzyme.

### 2.2. BSA Digestion with Modified Silicas

Once it was demonstrated that these materials were capable of immobilizing the Alcalase enzyme, the next step was to optimize the conditions promoting the enzyme activity. For that purpose, the probe reaction was the digestion of BSA, which was monitored by evaluating the amount of released peptides. According to enzyme specifications, Alcalase is expected to show its highest activity at pH 7–9 and temperatures from 45–65 °C. Previous works of our research group have studied the effect of pH on free Alcalase activity [43], observing that pH 8.5 resulted in the highest enzyme activity. Thus, this pH was employed for studying the influence of the following parameters on the digestion of BSA: temperature (50, 60, or 70 °C), time (2 or 4 h), amount of support immobilizing Alcalase (2.5 or 5.0 mg), and stirring speed during the digestion (700, 850, or 1000 rpm).

Figure 5 shows the results obtained when using Alcalase immobilized on **2-SiO_2_ (2-SiO_2_-Alc)**, and **3-SiO_2_ (3-SiO_2_-Alc)**. As expected, more peptides were released when using a higher amount of support (Figure 5A). No significant effect on the amount of released peptides was observed when increasing the agitation speed (Figure 5B). Moreover, Figure 5A,5B also show that the amount of released peptides was higher when the enzyme was immobilized on **2-SiO_2_-Alc**, which is in agreement with the results in Figure 3. Furthermore, the immobilization of the enzyme enabled an increase in its stability at high temperatures, as observed in Figure 5C. Indeed, whereas free Alcalase showed a significant reduced activity at a temperature of 70 °C, the immobilized Alcalase kept its activity better at this temperature. On the other hand, a reaction time higher than 2 h did not result in an increasing release of peptides when Alcalase was immobilized on **2-SiO_2_-Alc** and **3-SiO_2_-Alc** (Figure 5D) (*p*-value > 0.05). This result was compared with the result obtained when using the enzyme immobilized in the support modified with the linear molecule (**1-SiO_2_-Alc**). The enzyme immobilized on the **1-SiO_2_-Alc** support showed no activity at 2 h, and required 4 h for the digestion of BSA. This fact is in agreement with the activity reported for Alcalase grafted directly onto a material surface with no multipoint linker [36]. Hence, the best conditions for the digestion of BSA using immobilized Alcalase were: 5.0 mg of support, 700 rpm, 60 °C, and 2 h of reaction time.

Next, the activity of the enzyme immobilized on the different X-SiO2-Alc supports (X = 1–6) on the digestion of BSA under optimized conditions (5.0 mg of support, 700 rpm, 60 °C, and 2 h of reaction time) was evaluated and compared. The results are included in Figure 6. Enzyme activity was lower when using CBS dendrimers, especially for the G1 dendrimer, and higher when using PAMAM dendrimers. Most likely, the hydrophilic character of the PAMAM framework promoted the exposure of the enzyme for the digestion of BSA in a water environment, unlike CBS dendrimers with a more hydrophobic framework. These results were in agreement with those presented in Figure 3. In view of these results, the next experiments were performed with the support **5-SiO2** modified with the G0-PAMAM dendrimer.

### 2.3. Optimization of Immobilization Conditions and Study of Its Effect on the Activity of the Enzyme

In order to evaluate the effect of immobilization conditions on the amount of immobilized enzyme and on its activity, different parameters were next studied using the selected G_0_-PAMAM support (**5-SiO_2_**): immobilization time, Alcalase/SiO_2_ ratio, and the presence of Ca^2+^ ions. Figure 7 shows the obtained results. The immobilization time did affect the amount of Alcalase grafted on silica (Figure 7A1), observing a higher immobilization at longer reaction times. However, a higher amount of immobilized Alcalase only slightly modified the enzyme activity (Figure 7A2). Therefore, an immobilization time of 1 h was selected.

Regarding the effect of the enzyme/SiO_2_ ratio on the activity of the enzyme, the results demonstrated that the incorporation of Alcalase was not improved when increasing ratios (Figure 7B), so a 10 mg enzyme/100 mg SiO_2_ ratio was chosen.

On the other hand, Alcalase contains four Ca^2+^ in its allosteric center. The release of these cations could result in decreasing enzyme activity [44]. In order to study the influence of the presence of this cation on the activity of the immobilized enzyme, enzyme immobilization was carried out in presence or absence of 0.1 M CaCl_2_. The presence of CaCl_2_ made the incorporation of Alcalase to the material difficult with respect to the material without CaCl_2_, decreasing the amount of the immobilized enzyme.

### 2.4. Evaluation of the Reproducibility of the Immobilization Procedure and Reusability of the Immobilized Enzyme

The reproducibility of the immobilization procedure and reusability of the enzyme immobilized on the **5-SiO_2_** support modified with G_0_-PAMAM under conditions previously optimized were next evaluated.

The reproducibility of enzyme activity can be affected by the immobilization procedure, especially when the material modification is carried out in a heterogeneous medium. To evaluate this fact, the synthesis of the **5-SiO_2_** support was repeated three times. Figure 8A shows no significant difference on the activity of each batch towards BSA hydrolysis, being in the range 2.6–3.0 mg/mL. This activity was a bit lower than that observed for the free Alcalase (4.2 mg/mL) and corresponded to up to 71% of the free enzyme activity. This is a consequence of immobilization and has been reported in the bibliography when grafting enzymes [8,34,45]. This fact has been ascribed to alteration of the enzyme conformation produced during the immobilization reaction and to diffusional limitations [46,47]. Noteworthy in this regard, Wang et al. [48] observed Michaelis–Menten constant (Km) values for Alcalase immobilized on chitosan and its free form of 12.8 and 10.6 mg/mL, respectively.

The reusability of the enzyme immobilized on the **5-SiO_2_** support was studied by its use in a second digestion of BSA. The results demonstrated that the immobilized enzyme was active, although its activity reduced in a range from 26–50% (Figure 8A). This behavior is common in immobilized enzymes and was previously observed in the immobilization of thermolysin in a similar support [34]. In this case, the activity reduction was from 43 to 58% after the second digestion. Moreover, similar reductions after digestion were also observed in the immobilization of Alcalase in other supports, such as agarose beads [39], chitosan [37,48], and polymer-coated mesoporous silica nanoparticles [49].

Trying to keep Alcalase activity after first BSA digestion, supports prepared at higher Alcalase/SiO_2_ ratios were employed. The results demonstrated that higher ratios of the Alcalase enzyme resulted in more significant reductions in enzyme activity after first BSA digestion (Figure 8B). This reduction was compared with that observed for the support modified with the linear molecule (**1-SiO_2_**), observing that its activity was reduced to zero after the first BSA digestion. Therefore, the modification of silica supports with dendrimers enables reusing the enzyme.

Different reasons could explain the reduction in enzyme activity after the first BSA digestion: the enzyme could be autolyzed under digestion conditions, or peptides released in BSA digestion could be retained on the support surface, blocking the enzyme.

To evaluate the possible autolysis of the immobilized enzyme, it was submitted to the conditions employed for the digestion of BSA, but in the absence of this protein. The evaluation of the presence of amino groups in the resulting supernatant showed that there were no amino groups and, thus, the autolysis of the enzyme was not taking place.

To find out whether peptides released in the BSA digestion were retained on the material surface, the **5-SiO_2_** support immobilizing the enzyme was hydrolyzed with hydrochloric acid before and after BSA digestion, and amino acids released were determined by HPLC. If support was not affected by BSA digestion, the amount of amino acids released after the hydrolysis of the support must be identical to that determined previously to BSA digestion. Figure 9 shows that there were no significant differences within amino acid contents, before and after digestion, demonstrating that peptides were not trapped on the **5-SiO_2_** support causing enzyme blocking.

To have a deeper insight into this behavior, the other supports used in this work were also submitted to the same hydrolysis before and after BSA digestion, and the results were compared to those obtained with the **5-SiO_2_** support. Supports modified with CBS dendrimers (**2-SiO_2_**, **3-SiO_2_**, and **4-SiO_2_**) and the support modified with the linear molecule (**1-SiO_2_**) (Figure 9) showed an increase in the amount of amino acids after BSA digestion, which is likely due to the retention of peptides released in BSA digestion. This increase was especially significant in the case of the support modified with the linear molecule (**1-SiO_2_**) and was in agreement with previous results, demonstrating no activity of the enzyme immobilized on the linear molecule after the first BSA digestion. Regarding CBS dendrimers, the **4-SiO_2_**, derived from the polyphenoxo core, was less prone to interact with peptides, probably due to the more open framework of the dendrimer. The affinity of the hydrophobic skeleton of CBS dendrimers toward hydrophobic regions of biomacromolecules could justify the different behavior of these supports in comparison with PAMAM dendrimers with a hydrophilic framework [33,50].

The retention of peptides released in BSA digestion on the supports modified with CBS dendrimers and linear molecules was also observed in the immobilization of thermolysin in the silicas modified with these ligands [34].

## 3. Materials and Methods

### 3.1. Chemicals and Samples

All reagents were of analytical grade, and water was obtained with a Milli-Q system from Millipore (Bedford, MA, USA). Acetonitrile (ACN) was from Scharlau (Barcelona, Spain), di-sodium tetraborate decahydrate (borate buffer) from Merck (Darmstadt, Germany), and standard amino acids (AA-18) from Fluka (Buchs, Switzerland). Glutaraldehyde, sodium borohydrate (NaHB4), sodium tetraborate, β-mercaptoethanol, o-phthaldialdehyde (OPA), L-glutathione (GSH), and bovine serum albumin (BSA) were acquired from Sigma-Aldrich (St Louis, MO, USA). Hydrochloric acid (HCl) and calcium chloride (CaCl2) were from Panreac (Darmstadt, Germany). Alcalase was kindly donated by Novozymes Spain S.A. (Madrid, Spain).

### 3.2. Immobilization of Alcalase

CBS dendrimers, GnX (G_0_ and G_1_), were synthesized as previously published [24]. Linear molecules (2-aminoethanethiol) and PAMAM (ethylendiamine) dendrimers (G_0_ and G_1_) were acquired from Sigma-Aldrich. CBS and PAMAM dendrimers and linear molecules were attached to the silica support following a method previously described [33]. Afterwards, supports were activated with glutaraldehyde and the enzyme was immobilized. The coating with glutaraldehyde was performed by mixing 50 mg of supports in 5 mM borate buffer (pH = 8.5) with excess glutaraldehyde (0.125 mL, 0.7 mmol, 5 mol per-NH2). The reaction was stirred for 24 h at 30 °C under dark conditions. Then, the suspension was centrifuged at 2000× *g* for 8 min and the pellet was washed twice with distilled water to get rid of the remaining glutaraldehyde. Next, immobilization of Alcalase was carried out by adding 5 mL of a 4 mg/mL enzyme solution in borate buffer (pH = 8.5) and stirring for 1 h at 30 °C in the dark. Then, the suspension was centrifuged at 2000× *g* for 8 min. The remaining solid was suspended in 1.25 mL NaBH4 (0.53 mmol) to reduce double bonds and suppress reactive groups. This reaction was stirred at room temperature for 30 min in dark conditions. Finally, the mixture was centrifuged (2000× *g*, 8 min) and washed three times with distilled water. Then, the solid was dried under vacuum and stored at 4 °C until use.

### 3.3. SEM Analysis

The morphology of silica supports coated with dendrimers before and after Alcalase immobilization was studied by scanning electron microscopy (SEM, Hitachi TM-1000, Tokyo, Japan).

### 3.4. Fluorescence Measurements

The amount of enzyme immobilized on supports was monitored by fluorescence spectroscopy using a RF-6000 spectro fluorophotometer from Shimadzu (Kyoto, Japan). The λexc was fixed at 280 and the λem ranged from 290 to 400 nm. The amount of immobilized enzyme on functionalized support was calculated as the difference between the fluorescence intensity corresponding to the free enzyme and the fluorescence intensity of the enzyme remaining in the supernatant after immobilization.

### 3.5. Amino Acid Analysis

The analysis of amino acids was carried out to demonstrate enzyme immobilization. For that purpose, the immobilized enzyme was hydrolyzed with 6 M HCl using a Milestone ETHOS D microwave oven from Gomensoro (Madrid, Spain) under argon conditions at 170 °C for 20 min. After digestion, samples were diluted up to 10 mL with Milli-Q water and filtered with nylon filters (13 mm diameter and 0.45 μm pore size) from Agilent Technologies (Palo Alto, CA, USA). Amino acids were analyzed by RP-HPLC previous derivatization with OPA using a Hypersil AA-ODS column (2.1 mm × 200 mm, 5 μm particle size) from Agilent Technologies. OPA derivatization was carried out in the injector by mixing 8 μL of OPA, 6 μL of borate buffer, 4 μL of sample, and 6 μL of borate buffer. The reaction took 2 min and immediately after, samples were analyzed by RP-HPLC. The chromatographic conditions were: mobile phase A, sodium acetate (50 mM, pH 7.2); mobile phase B, ACN; injection volume, 24 μL; flow rate, 0.5 mL/min; column temperature, 20 °C; elution gradient, 0–50% B for 40 min, 50–90% B for 5 min, and 90% B for 5 min. Fluorescence detection was recorded at a λexc of 340 nm and a λem of 450 nm. Amino acid identification was carried out by comparison with amino acid standards.

### 3.6. Enzyme Activity Assay

The digestion of BSA standard protein using free or immobilized Alcalase was carried out to evaluate the activity of the enzyme. For that purpose, 5 mg of BSA was dissolved in borate buffer (5 mM, pH 8.5) at a concentration of 5 mg/mL. This solution was mixed with 5 mg of immobilized Alcalase or with the free enzyme at a ratio of 0.15 UA/g BSA. Hydrolysis was carried out by incubation in a Thermomixer Compact (Eppendorf AG, Hamburg, Germany) for 2 h at 60 °C and 700 rpm. The reaction was stopped by increasing the temperature to 100 °C for 10 min followed by centrifugation (10 min, 6000 rpm) when using the free enzyme, or directly by centrifugation (10 min, 6000 rpm) when using the immobilized enzyme. Supernatants containing peptides were collected and stored at –20 °C. The pellet containing the immobilized enzyme, obtained after centrifugation, was washed for the reutilization of the enzyme. The activity of the free and immobilized Alcalase was measured by monitoring the released peptides by the OPA method [51].

### 3.7. Statistical Analysis

Statistical analysis was performed using Statgraphics Software Plus 5.1 (Statpoint Technologies, Inc., Warranton, VA, USA). Values were expressed as mean ± standard deviation of at least three independent experiments. The analysis of variance (ANOVA) was performed using a significant level of 0.05.

## 4. Conclusions

The multivalency of dendrimers is useful to attach and stabilize enzymes such as Alcalase on silica surfaces. The enzyme immobilized on supports modified with dendrimers could be used at higher temperature and required shorter digestion times than the free enzyme or the enzyme immobilized on the support modified with a linear molecule. The activity of the immobilized enzyme highly depended on the kind of dendrimer and its generation. PAMAM dendrimers promoted the immobilization and the activity of the enzyme, most likely due to its hydrophilic character, which supported the exposure of the enzyme in a water environment, whereas CBS dendrimers, with a more hydrophobic framework, were less favorable. The activity of the immobilized enzyme was lower in supports modified with G_1_ dendrimers than in G_0_, likely due to the increased rigidity of the enzyme. Alkyl chains and CBS dendrimers can more easily retain peptides released in BSA digestion, due to their hydrophobic core, compared to more polar systems such as PAMAM. The support modified with the G_0_-PAMAM dendrimer was more favorable for successive digestions than supports modified with CBS dendrimers or the monofunctional molecule.

## Figures and Tables

**Figure 1 ijms-23-16102-f001:**
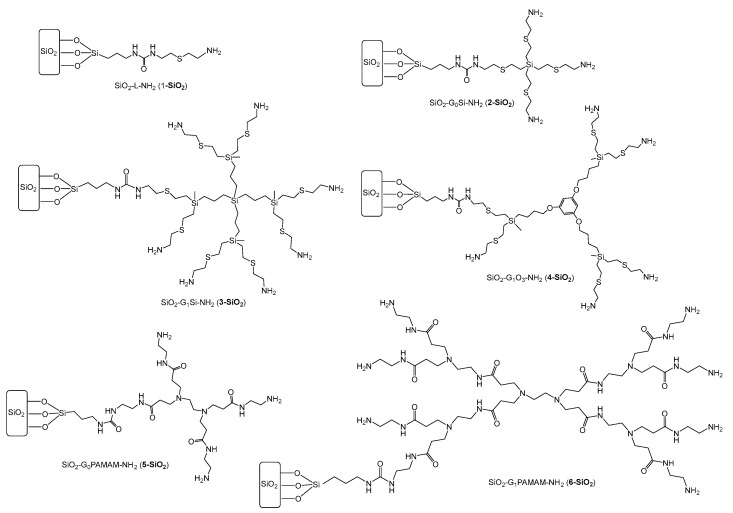
Drawing of silica modified with CBS and PAMAM dendrimers and monofunctional model.

**Figure 2 ijms-23-16102-f002:**
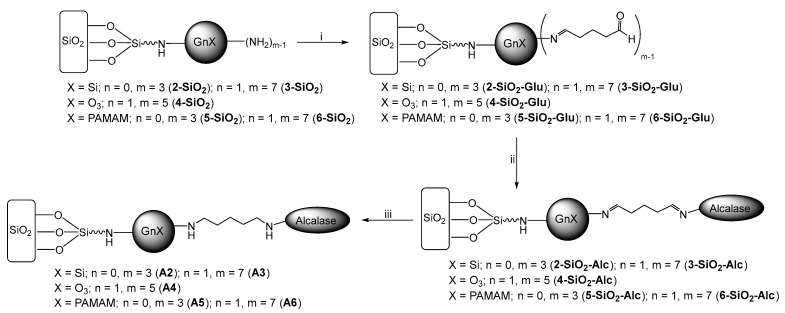
Immobilization of Alcalase over silica surface modified with dendrimers: (**i**) activation with glutaraldehyde; (**ii**) immobilization of the enzyme; (**iii**) reduction of double bonds.

**Figure 3 ijms-23-16102-f003:**
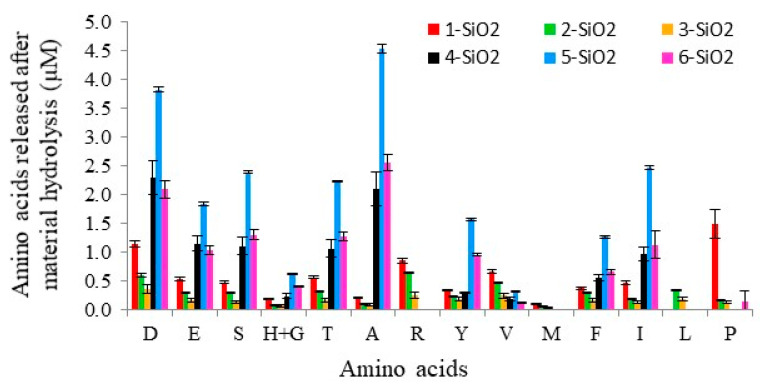
Analysis of amino acids (**D**: aspartic acid, **E**: glutamic acid, **S**: serine, **H**: histidine, **G**: glycine, **T**: threonine, **A**: alanine, **R**: arginine, **Y**: tyrosine, **V**: valine, **M**: methionine, **F**: phenylalanine, **I**: isoleucine, **L**: leucine, **P**: proline) released in the hydrolysis with hydrochloric acid of Alcalase immobilized on different supports.

**Figure 4 ijms-23-16102-f004:**
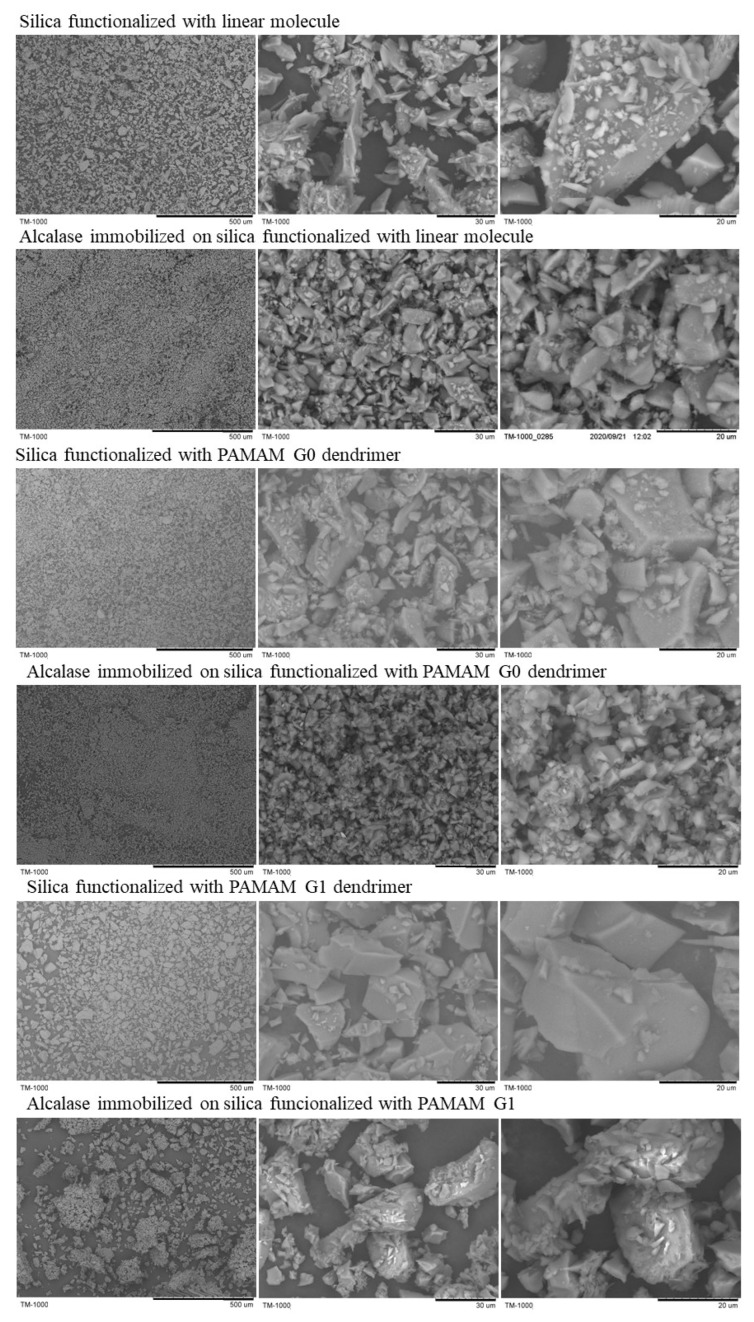
Scanning electron microscopy (SEM) images at different resolutions of the silica supports functionalized with the linear molecule and the PAMAM dendrimers (G0 and G1) (obtained from reference [34]) and of the functionalized silica supports immobilizing Alcalase.

**Figure 5 ijms-23-16102-f005:**
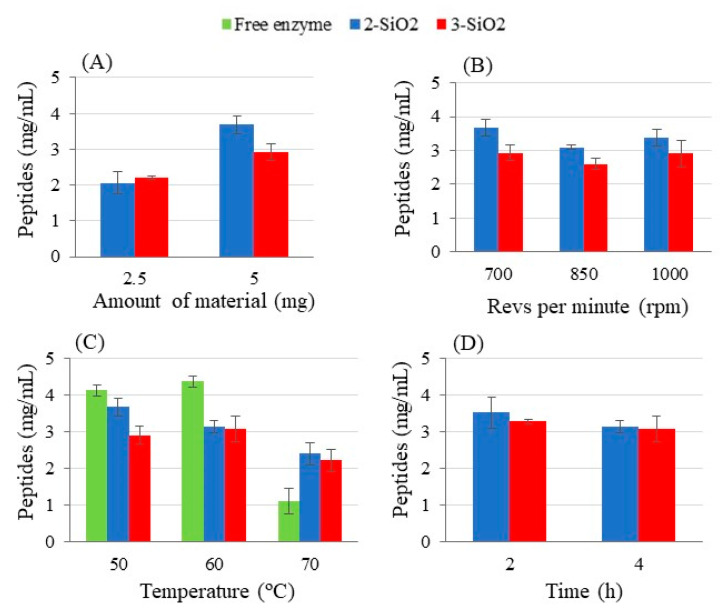
Effect of the amount of material (**A**), stirring speed (**B**), temperature (**C**), and time (**D**) on the digestion of BSA using Alcalase immobilized on 2-SiO_2_ and 3-SiO_2_ supports. Other conditions were: (**A**) 700 rpm, 50 °C, and 4 h; (**B**) 5 mg material, 50 °C, and 4 h; (**C**) 5 mg material, 700 rpm, and 4 h; (**D**) 5 mg material, 700 rpm, and 60 °C.

**Figure 6 ijms-23-16102-f006:**
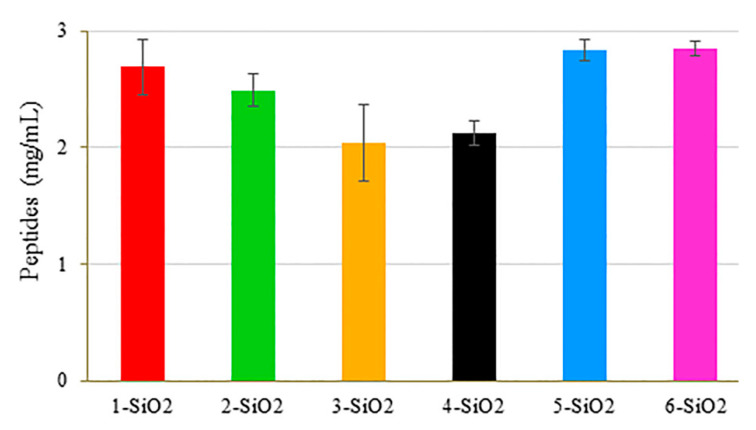
Amount of peptides released in the digestion of BSA using Alcalase immobilized on different grafted system.

**Figure 7 ijms-23-16102-f007:**
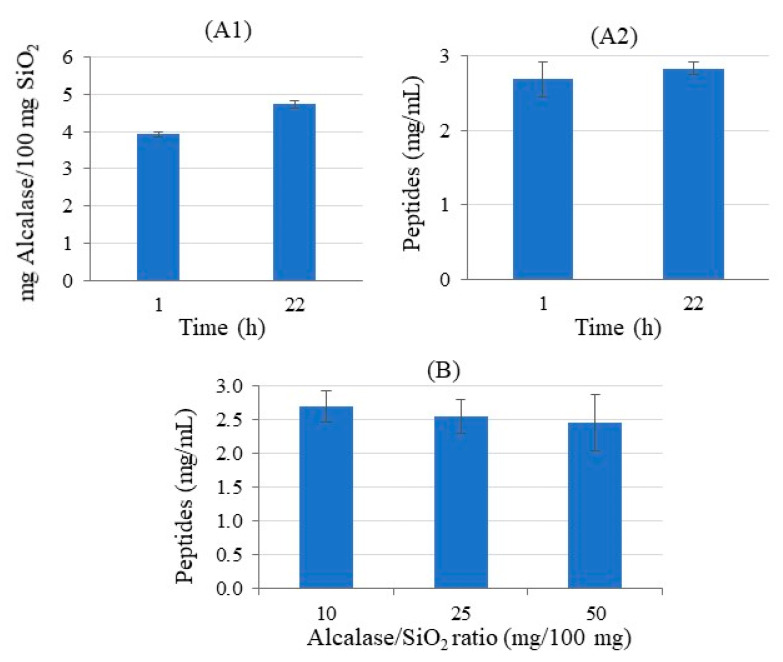
Amount of immobilized Alcalase per 100 mg SiO_2_ regarding to the immobilization time (**A1**) and activity of the immobilized enzyme in BSA digestion after varying different immobilization conditions: (**A2**) reaction time and (**B**) Alcalase/SiO_2_ ratio.

**Figure 8 ijms-23-16102-f008:**
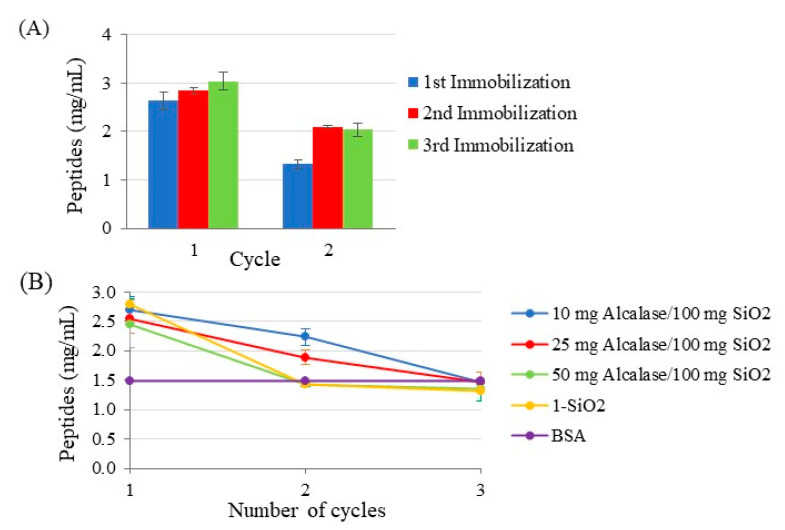
Evaluation of the reproducibility (**A**) and reusability (**B**) of Alcalase immobilized on 5-SiO_2_ support.

**Figure 9 ijms-23-16102-f009:**
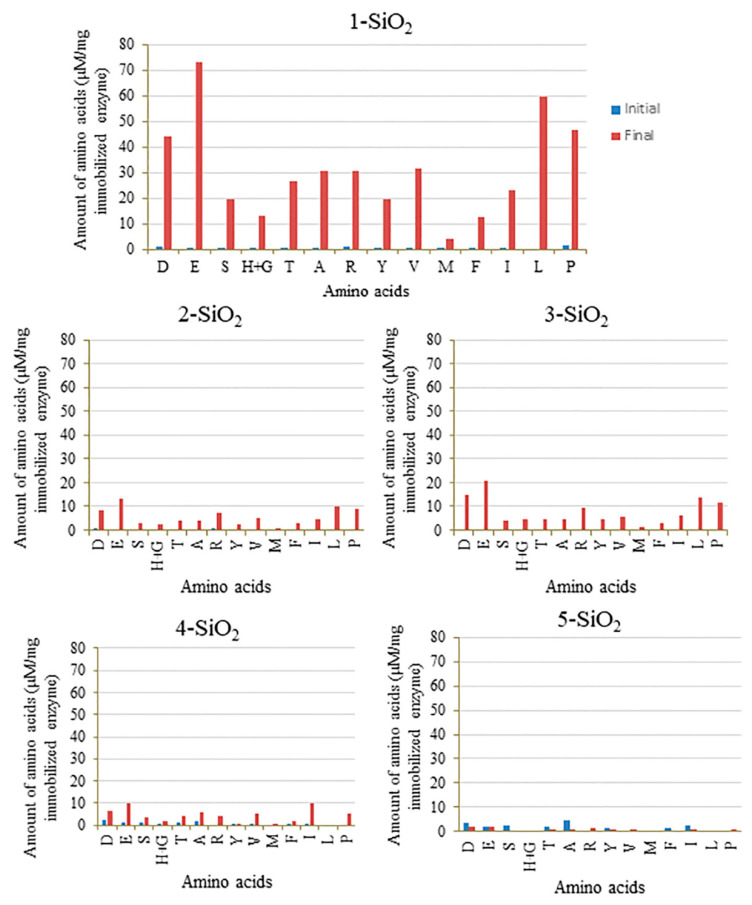
Comparison of amino acids released in the hydrolysis of Alcalase immobilized on different supports, before and after BSA digestion.

## Data Availability

All data generated or analyzed during the study are included within the article.

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
