# Peer review of "Immobilization of Alcalase on Silica Supports Modified with Carbosilane and PAMAM Dendrimers"

_ijms, 2022, doi:10.3390/ijms232416102_

Round 1
Reviewer 1 Report
The authors demsotrated immobilize Alcalase on silica using different den-81 drimer-based supports: CBS, with hydrophobic framework, and polyamidoamine (PA-82 MAM), with hydrophilic framework, and the work is intresting and can be accepted after include some SEM images in the main articles. Otherwise work is perfect to publish.
Author Response
Following reviewer suggestion, SEM images were added to the main article in the new manuscript (Figure 4).
Reviewer 2 Report
The article « Immobilization of Alcalase on silica supports modified with 2 carbosilane and PAMAM dendrimers» is attracted to a relevant topic and has a high applied value.
The authors present five different dendrimers supported on silica to immobilize Alcalase. Results were compared with a silica support modified with a monofunctional molecule (2-aminoethanethiol). The effect of the dendrimer generation, the immobilization conditions (immobilization time, alcalase/SiO2 ratio, and presence of Ca2+ ions), and the temperature, time, amount of support, and stirring speed on Alcalase activity has been evaluated.
The article is well structured, written in sufficient detail and logically. The authors conducted an extensive experiment.
Major comments:
1. Authors should submit рН and temperature profile and long-term рН and thermostability profile of immobilized Alcalase and Alcalase in solution.
2. The study should be supplemented by the determination of catalytic constants, such as Km, Vmax and Kcat.
Minor remark:
Line 75: Bacillus licheniformis should be italicized
Author Response
- Authors should submit рН and temperature profile and long-term рНand thermostability profile of immobilized Alcalase and Alcalase in solution.
Alcalase enzyme is stable from pH 7 to 9 and at temperatures from 45 to 65 ºC. Moreover, previous works of our research group have studied the effect of pH on the activity of free Alcalase (J. Funct. Foods, 11 (2014) 428) observing that pH 8.5 resulted in the highest enzyme activity. Taking into account this information, the selected pH was 8.5. This information was added in the lines 156-158 of the new manuscript to clarify the use of this pH. Regarding temperature, this work studied the effect of this parameter on the stability of the enzyme and results were showed in Figure 5C. As observed in this Figure, the activity of Alcalase in solution significantly decreased at 70 ºC, while the immobilized Alcalase stood better this high temperature. This information has been highlighted in the new version of the manuscript, to avoid misunderstandings (lines 173-176).
- The study should be supplemented by the determination of catalytic constants, such as Km, Vmax and Kcat.
Thank you very much for this suggestion. Kinetic studies are very important and are included in our future pipeline. Nonetheless, we think that we will have similar results to the obtained in a previous work of our research group devoted to the immobilization of another enzyme, thermolysin. This research showed no significant differences in Vmax and Kcat between the immobilized enzyme and the free enzyme (Int. J. Biol. Macromol., 165 (2020), 2338). Moreover, Michaelis constant (Km) enabled to observe that the free enzyme presented higher affinity to the protein than immobilized enzyme and, as consequence, the catalytic efficiency ratio (Kcat/Km) confirmed that the free enzyme was more efficient to hydrolyze proteins than the immobilized enzyme. This was attributed to diffusional limitations, which took place when the hydrolysis was performed with the immobilized enzyme.
Minor remark:
Line 75: Bacillus licheniformis should be italicized
According to reviewer suggestion, Bacillus licheniformis was changed to italics in the new manuscript.
Round 2
Reviewer 2 Report
The authors' response to my question 1 satisfies me completely.
But I do not agree with the answer to the question 2, the kinetic parameters of different enzymes are different and their change after immobilization can differ significantly.
The authors should either carry out additional experiments to determine Km, Vmax and Kcat, or more convincingly justify the uselessness of their conduct, not only in the response to the reviewer, but also in the text of the article.
Author Response
Thank you very much for your suggestion. Results reported in this work showed that the free Alcalase had higher affinity towards the protein than the immobilized Alcalase. This was attributed to alteration of enzyme conformation produced during the immobilization reaction and to diffusional limitations. This has also been reported in other works immobilizing Alcalase (Am. J. Food Technol. 2016, 11, 152; Eur. Food Res. Technol. 2014, 239, 1051) or other enzymes (Biotechnol. Prog. 2003, 19, 352; Int. J. Biol. Macromol. 2020, 165, 2338; Bull Korean Chem. Soc. 2012, 33, 2181; Nanoscale. 2016, 8, 6728). Moreover, kinetic studies have supported these results. Indeed, a previous study of our research group using thermolysin enzyme demonstrated that BSA hydrolysis with free enzyme was more effective than that using the immobilized enzyme. Moreover, Michaelis constant (Km), corresponding to the substrate concentration at half of Vmax and related to the affinity of enzyme towards the substrate, was higher when using the immobilized enzyme than with the free one (Km immobilized = 1.96 mg/mL and Km free = 1.4 mg/mL) (Int. J. Biol. Macromol. 2020, 165, 2338). Wang et al. (Eur. Food Res. Technol. 2014, 239, 1051) also observed Km values for the free Alcalase lower than the corresponding to the immobilized Alcalase on chitosan (10.6 ± 0.01 and 12.8 ± 0.01 mg/mL, respectively). Similarly, laccase enzyme immobilized on yolk-shell particles showed a Km = 41.5 μM, while the free enzyme had a Km = 29.3 μM (Nanoscale. 2016, 8, 6728). These results support our results. According to reviewer suggestion, this information was added to the new manuscript (lines 230-243).